# More Than 200 Years Later: *Gluvia brunnea* sp. nov. (Solifugae, Daesiidae), a Second Species of Camel Spider from the Iberian Peninsula

**DOI:** 10.3390/insects15040284

**Published:** 2024-04-17

**Authors:** Cristian Pertegal, Pablo Barranco, Eva De Mas, Jordi Moya-Laraño

**Affiliations:** 1Departamento de Ecología Funcional y Evolutiva, Estación Experimental de Zonas Áridas, (EEZA-CSIC) Ctra. de Sacramento s/n, La Cañada de San Urbano, 04120 Almería, Spain; demas@eeza.csic.es (E.D.M.); jordi@eeza.csic.es (J.M.-L.); 2Departamento de Zoología, Universidad de Córdoba, Edificio C-1, Campus de Rabanales, 14071 Córdoba, Spain; 3Departamento de Biología y Geología, CITE-IIB, Centro de Colecciones, CECOUAL, Universidad de Almería, Ctra. Sacramento, s/n, La Cañada, 04120 Almería, Spain; pbvega@ual.es

**Keywords:** Daesiidae, *Gluvia*, Iberian Peninsula, new species, Solifugae, taxonomy

## Abstract

**Simple Summary:**

The description of a new species of camel spider, *Gluvia brunnea* sp. nov. Thorough morphological details, as well as molecular DNA and statistical analyses, serves to clearly differentiate this new species from *Gluvia dorsalis* (Latreille, 1817), the only species previously known from the Iberian Peninsula.

**Abstract:**

We present the description of a new species of Solifugae from the Iberian Peninsula, *Gluvia brunnea* sp. nov., which has been found so far in southeast Spain. The morphological description is accompanied by molecular and multiple factor analyses, jointly giving full support to the specific status of the taxon. Finally, we discuss the intraspecific variability of both species, *G. dorsalis* and *G. brunnea* sp. nov., and the recent history of the genus. We also discuss the usefulness of multiple factor analysis for quantitatively separating species, and we stress that some specimens of this new species were found in Mesovoid Shallow Substratum stations, representing the very first time that Solifugae have been captured in this type of trap.

## 1. Introduction

The scientific interest in the order Solifugae has grown in the last few years, beginning to close a long-standing gap in knowledge on the biology of these animals [1,2,3,4]. Additionally, molecular DNA analysis has helped us to determine the systematics of the group, e.g., the recent reassignation of previous subfamilies to the new families Dinorhaxidae [5], Lipophagidae [6] and Namibesiidae [6,7]. Currently, the order includes 15 families, 144 genera and 1209 species, according to the World Solifugae Catalogue [8]. More specifically, for the Iberian Peninsula, a single endemic species has been known, namely *Gluvia dorsalis* of the family Daesiidae Karepelin, 1899 [7,9].

The taxonomy and systematics of Solifugae have always posed a challenge [1,2,3,10,11], particularly due to the lack of consensus among specialists about the relevant morphological characteristics. Adding to this problem, some of the candidate traits show relatively high intraspecific variation [12]. Fortunately, the relatively recent revision of the cheliceral morphology by Bird et al. (2015) [1] provided a better understanding of this order, thus enabling the accurate identification and description of Solifugae species. As for the family Daesiidae, subfamily-level divisions have been problematic due to inconsistencies in morphology and biogeography [3,6,13]. The genus *Gluvia* [14] is one of the 23 genera belonging to this family. After several reviews on the taxonomy of this genus, the number of species was reduced to one following transfers of some congeners to other genera and following the recognition that the descriptions of most remaining species were based on unreliable characteristics, e.g., *G. atlantica*, [15] (*nomen dubium*) from Cabo Verde [5,10,15,16,17,18,19,20].

To help overcome the types of difficulties sometimes posed by morphological data, DNA analysis may be a useful complementary tool. Barcoding based on the mitochondrial gene encoding the enzyme cytochrome b oxidase (COI) has been widely used for identifying species [21], and barcoding has been proven useful to confirm the taxonomy based on morphology. There are examples of COI use for most of the arachnid taxa (e.g., Ricinulei [22], Acarina [23], Pseudoscorpionida [24], Opiliones [25], Araneae [26]), and combined with morphological characteristics (i.e., integrative taxonomy), it has become an invaluable tool for diversity assessment and to solve taxonomic inconsistencies [27,28,29,30,31].

In the present work, we describe a new species in the genus *Gluvia* from the southeast of the Iberian Peninsula. To this end, we integrate a more traditional morphological description of characteristics, with statistical analysis of continuous, count and categorical descriptors (multiple factor analysis), along with DNA (mitochondrial gene COI) analysis. In these analyses, we compare specimens of the proposed species with others of *G. dorsalis*, which were collected from several localities in the Iberian Peninsula.

## 2. Material and Methods

### 2.1. Sampling Methods

*G. dorsalis* specimens were sourced through direct capture or pitfall traps set by either colleagues or ourselves. Specimens of the new species came from the following three studies: Two specimens from Sierra de María, Almería (Figure 1), were collected in the course of a study of the interstitial entomofauna (Canchal Maimón-II, Latitude: 37.66347, Longitude: 2.11141, 1230 m.a.s.l.), through the installing of Mesovoid Shallow Substratum stations in colluvial stony debris in the eastern sector of the mountain range, following the usual method for these studies [32,33,34,35,36,37]. These stations consist of a PVC cylinder of 1 m in length X 11 cm in diameter, with several 8 mm diameter holes in the lower section, so that, after being installed vertically in the subsoil, fauna can enter at depths of 0.5 to 0.9 m. A pitfall trap containing 1,2-propanodyol as a preservative and odorous cheese as bait is carefully slid to the bottom of the cylinder. They were visited every three months to obtain pitfall samples from July 2018 to December 2019 and, at the moment of collection, 50 mL of 96% alcohol was available in situ to preserve the sample. Furthermore, specimens from the Sierra de Gádor, Almería, were collected on the perimeter of the Castala Periurban Park, Berja, Almería, to study how the entomofauna was affected in a burned area. For that study, a total of 72 pitfall traps were used. Each trap consisted of a 7.5 cm diameter and 200 mL plastic container filled with 100 mL of propylene glycol and buried at the soil level with the top edge of the container on the surface. The pitfalls were active between June and July 2021, with 24 traps (3 traps × 8 plots) collected in a total of three visits. At the time of collection, 20 mL of 96% ethanol was added to each sample container to preserve the samples until their processing and separation. Additionally, another set of specimens was sourced from an ongoing experiment (project Spill-Island) in Cabo de Gata Natural Park (Almería), comprising a Biosphere Preserve in a sub-desert area where rainfall is the lowest in continental Europe [38]. As part of this experiment, 20 × 20 cm square-section pitfall traps were set on different dates to study the activity (live traps) and diversity (propylene glycol) of macrofauna in 30 plots scattered across the park. The main vegetation in the sampling areas are dwarf palms (*Chamaerops humilis*) and *Stipa capensis* grass. All *Gluvia* specimens from this last study were collected during the months of June and July of either 2021 or 2022 (Figure 1). Finally, other samples from different localities were collected by hand when searching during night transects with a head lamp (Figure 1).

### 2.2. Scanning Electron Microscopy

SEM (scanning electron microscopy) was used to visualize the six specimens: two pairs (female and male) of *G. dorsalis*, one from Córdoba and one from Huelva, and one pair of the new species. Their different structures were dehydrated and metallized with a 10 nm gold coating (LEICA EM 200, Wetzlar, Germany) for electron microscopic observation (Sigma 300VP High Field Resolution, Jena, Germany) at a high vacuum.

### 2.3. Morphological Characterization

The morphological description of the present work follows the directions of previous authors [1,39], plus other complementary characteristics that could be of importance, at least for this taxon, such as interocular measurement, ocular tubercle area, propeltidium area, flatness ratio of the propeltidium, flagellum length, flagellum width at the insertion point and width of the flagellum flag (Figure 2). A notch in the flagellum separates a proximal (flag) from a distal part. This notch is indicated in the photographs as “n”. All measurements were carried out on specimens preserved in alcohol. Microdissection scissors and a hypodermic needle were needed to dissect the specimens to characterize the prolateral area of chelicerae and to take photographs in SEM. Measurements were taken using a Hayear 5130 ocular camera coupled with a Euromex SB. 1903-P stereomicroscope, and photographs were taken with the specimens immersed in 70% EtoH using an E3ISPM 20000 camera coupled with the same stereomicroscope.

Abbreviations: Collections: CECOUAL: Centro de Investigación de Colecciones Científicas de la Universidad de Almería; MNCN: Museo Nacional de Ciencias Naturales; EEZA: Estación Experimental de Zonas Áridas. Tooth position (following [1]): FD: fixed finger distal tooth; FSD: fixed finger subdistal tooth; FM: fixed finger medial tooth; FP: fixed finger proximal tooth; RFD: retrofondal distal tooth; RFM: retrofondal medial tooth; RFSM: retrofondal submedial tooth; RFSP: retrofondal subproximal tooth; RFP: retrofondal proximal tooth; PFD: profondal distal tooth, PFM: profondal medial tooth; PFP: profondal proximal tooth; MM: movable finger medial tooth; MP: movable finger proximal tooth; MSM: movable finger submedial tooth.

### 2.4. Statistical Analyses

Multiple factor analysis (MFA)—In order to test whether the traits measured contributed to multidimensional separation between *Gluvia* species, continuous, count and qualitative characteristics were jointly analyzed in two multiple factor analyses [40], one for each sex. MFAs were run using the R library “FactoMineR” [41]. This technique, unlike some others commonly used (e.g., discriminant analysis), can handle a smaller number of cases than variables. However, in order to minimize redundancy in the MFA, highly correlated continuous measurements, corresponding to some morphological traits common in both sexes, were automatically detected and discarded by using the R function *findCorrelation* from the *caret* package [42]. The Pearson correlation cutoff set for variable removal was 0.8. This means that variables that were too correlated with the rest (e.g., teleotarsus II) were not considered for analysis. The continuous variables that were finally included were distributed in ordination groups as follows: *Propeltidium*, flatness_ratio and width; *Eyes*, eye length; *Pedipalps*, teleotarsus and femur length; *Legs*, teleotarsus I, basitarsus I, tibia I, femur I, claw I, basitarsus II, femur II, claw II, teleotarsus III, femur III, claw III and total leg IV length; *Flagellum* (only males), length and width and insertion point. These last variables were centered on the unit variance for analysis. The number of structures varying in number (count variable) was included in groups of continuous variables but without scaling each to mean zero and unit variance. These groups and variables were *Chelicerae*, FD, FP, PFD, RFD+RFM+RFSM+RFSP and PFD+PFM+PFP; *Stridulatory organ*, number of ridges in the stridulatory plate, number of setae rows, number of filiform setae and number of plumous setae. General coloration (both males and females) and the presence or absence of a “skirt-like row of dense hair” in males were included as qualitative variables. Species identity was included as a supplementary qualitative variable, which served to obtain the centroids and 95% confidence ellipses of each species on the ordination space, but did not contribute to the ordination itself. The function *fviz_mfa_ind* in library *factoextra* [43] was used to plot the results. Finally, to test if MFA successfully separated the two species, the score values for the two first dimensions (explaining a large proportion of the overall variance) were tested for overall significance in a multivariate analysis of variance (MANOVA) including the scores as dependent variables and the identity of species as an independent categorical variable (function “manova” in the R library “stats”). The assumptions of univariate and multivariate normality, heteroscedasticity and homogeneity of covariances were tested using the libraries “MVN” [44] (multi- and uni-normality), “car” [45] (Levene’s test for homogeneity of variances) and “biotools” [46] (Box M test for homogeneity of covariances). R version 4.3.4 was used for all analyses [47].

### 2.5. DNA Analysis

Barcoding species delimitation was used here to complement morphological differences. DNA barcoding was based on a 430 bp fragment of mitochondrial cytochrome oxidase subunit I (COI), which was analyzed in 15 Solifugae specimens. Mitochondrial DNA from leg I of each specimen was extracted using the DNeasy^®^ Blood & Tissue Kit of Qiagen. The COI fragment was amplified by PCR using forward primer III-B-F [48] and reverse primer Fol-degen-R [49]. The cycling conditions were 94 °C for 4 min and 40 cycles of 94 °C for 30 s—48 °C for 30 s—72 °C for 45 s—72 °C for 10 min and, finally, 10 °C to maintenance. Quantification of the PCR product was carried out by means of a Qubit^®^ fluorescence detector (Invitrogen, USA). A purified and sequencing reaction in both directions was performed by Macrogen Spain. The sequences were deposited in GenBank (PP229911–PP2299125). Sequence analysis was performed with molecular data from 18 individuals: 15 individual sequences were extracted and sequenced for the present study, and 3 sequences were obtained from GenBank (Table 1).

Available GenBank sequences of three different Solifugae species were added to the molecular analysis: two species of the Daesiidae family (*Gluvia dorsalis* MD038 and *Gluviopsis nigrocinctus*-Birula 1905-SOL1062) and one species of the Eremobatidae (Kraeplin 1899) family (*Eremobates scopulatellus*-Muma & Brookhart 1988-DMNS:ZA19206). The sequence of the mite *Galumna dimorpha* (Lacasa 1952) (Oribatida order) was also obtained for this study (extraction from the whole individual) and included as an outgroup (access sequence PP228019). We used GENEIOUS Prime^®^ 2023 1.1 for the editing and revision of sequences. Sequences assembled for consensus were built “de novo” for each individual within GENEIOUS. The alignment of individuals and the tree of genetic distances (in the form of a preliminary phylogenetic analysis) were completed with MEGA 11 [50]. For sequence alignment, we applied the muscle algorithm [51], and genetic distances were calculated using the Kimura 2-parameter method [52] with default parameter values. The tree of genetic distances across 19 individuals was built using the neighbor-joining (NJ) method [53]. Bootstrap support values were obtained by a bootstrap test [54] with 500 replicates.

## 3. Results

### 3.1. Taxonomy

*Gluvia brunnea* sp. nov. Pertegal, Barranco, De Mas & Moya-Laraño.

urn:lsid:zoobank.org:pub:89EF087C-9ECF-42F8-A8C0-F3BF5CDC8063

*Holotype:* Male, Fuente del Cerezo, T.M. Berja, Sierra de Gádor, Almería, Spain, CECOUAL leg., 21 June 2021 (longitude: −2.9167199, latitude: 36.8941099; 1043 m.a.s.l.), MNCN 20.02/38113.

*Allotype:* Female, esparto bush Mirador del Zarzalón, T.M. Berja, Sierra de Gádor, Almería, Spain, CECOUAL leg., 10 July 2021 (−2.888245, 36,8795417; 1507), MNCN 20.02/38114.

*Paratypes:* Twelve females and eight males as follows: one male, one female (SEM), pine forest repopulation Lavadero Mineral, 21 June 2021 (−2.9018983, 36.8988; 1327), (CECOUAL type numbers 0071, 0076). Two males, pine forest repopulation Fuente del Cerezo, 28 June 2021 (−2.9167199, 36.8941099; 1043), (CECOUAL type numbers 0072, 0073). Four males, holm oak forest Corraliza de los González, 28 June 2021 (−2.88191, 36.8714067; 1555), (CECOUAL type numbers 0074, 0075 (SEM); MNCN 20.02/38115, EEZA type number 668-1). Five females, pine forest repopulation Fuente del Cerezo, 10 July 2021 (−2.9198527, 36.893239; 1040), (CECOUAL type numbers 0062, 0063, 0064, 0065, 0077 (SEM)). One male, pine forest repopulation Lavadero Mineral, 10 July 2021 (−2.9018983, 36.8988; 1327), (CECOUAL type number 0070). Two females, holm oak forest Corraliza de los González, 10 July 2021 (−2.88191, 36.8714067; 1555), (MNCN 20.02/38116, EEZA type number 667-1). One female, holm oak forest Corraliza de los González, 11 July 2021 (−2.88191, 36.8714067; 1555), (CECOUAL type number 0066). Three females, pine forest repopulation Fuente del Cerezo, 12 July 2021 (−2.9198527, 36.893239; 1040), (CECOUAL type numbers 0067, 0068, 0069). All T.M. Berja, Sierra de Gádor, Almería, Spain, CECOUAL leg.

All specimens preserved in 70% alcohol, except those metallized for the SEM study, which were preserved dry glued on labels.

### 3.2. Etymology

The specific name is given due to the characteristic brown body color of this species, most obvious in living specimens with a dorsal view (Figure 3).

### 3.3. Diagnosis

*Gluvia dorsalis* and *Gluvia brunnea* sp. nov. can easily be distinguished by their colors. The first species has yellow areas in the palps and legs (Figure 3B,D), while the new species is completely brown in its dorsal view (Figure 3A,C), with the yellow color restricted to the ventral areas. In addition, there are further distinguishing characteristics in their morphology. Mature specimens of *G. brunnea* sp. nov. bear a hypertrophied seta on the basal and internal part of coxa I, pointing forward, which is absent in *G. dorsalis* (Figure 3E,F). The interocular distance in *G. dorsalis* is as small as half the length of the eyes (Figure 3H) while, in *G. brunnea* sp. nov., it is longer (Figure 3G). Other diagnostic characteristics are found only in adult males, such as the morphology of the flagellum, the morphology and number of teeth of the fixed finger, the mucron of the movable finger and the presence or absence of a ventral row of bristles on the fourth opisthosomal tergite similar to a skirt. The flagellum of *G. brunnea* sp. nov. is elevated (convex) in its dorsal edge, with a discontinuity (notch) in the middle of the ventral part delimitating a distal and a proximal lobe, the latter with a fringe edge (Figure 3I), while in *G. dorsalis*, the dorsal edge of the flagellum is smooth, the discontinuity is in the posterior first third of its length and the fringe edge of the smaller proximate lobe has longer extensions (Figure 3J). The fixed finger of *G. brunnea* sp. nov. has a sharpening mucron with soft edges and two distal teeth (well-developed FD and median-sized secondary teeth; Figure 3I), in contrast to the fixed finger of *G. dorsalis*, which has a robust mucron and one small distal tooth (small-sized FD; Figure 3J). The mucron of the movable finger is sharp and narrow at the base in *G. brunnea* sp. nov. (Figure 3I), but basally thicker in *G. dorsalis* (Figure 3J). Finally, a skirt-like row of hairs with a different density in the distal edge of the fourth sternite is present in *G. dorsalis* males (Figure 3L), but not in *G. brunnea* sp. nov. males (Figure 3K).

### 3.4. Description Holotype (Male)

Measurements as in Table 2.

Color. Preserved specimen brown in dorsal view (Figure 4A) and yellowish in ventral view (Figure 4B). Propeltidium dark brown with the anterior edge black and a longitudinal yellowish stripe, ocular tubercle blackish divided by a yellowish line (Figure 4C). Mesopeltidium, metapeltidium and opisthosoma dark brown. Chelicerae dorsally dark brown with yellowish lateral sides, stridulatory plate yellow, reddish orange fingers and black mucron (Figure 4D). Palps and legs brown with yellow ventral side. 

Prosoma. Propeltidium subcircular, wider than longer (Table 2), covered with hair and forked setae of different size (Figure 4C). Ocular tubercle of hexagonal shape, slightly elevated with two small apophyses in the anterior edge and rounded with abundant forked setae. Interocular distance a bit shorter than the length of the eye (Table 2). Propeltidial lobes separated from the propeltidium by a lateral shallow groove. Mesopeltidium and metapeltidium of rectangular shape and covered with forked setae (Figure 4C). Coxae also with abundant forked setae.

Chelicera-dentition and processes. Fixed finger dentition composed by two FD, one well-developed distal tooth and other smaller secondary tooth, one FSD with similar size of big FD; two medial teeth, FM bigger than FSM; one proximal tooth, FP; five retrofondal (RFD, RFM, RFSM, RFSP, RFP) teeth and three profondal teeth (PFD, PFM, PFP); sinuous somewhat elevated asetose area edge without dorsal crest; flat ventrodistal area and hardly visible lateral carinae. Fixed finger mucron with hook-shaped FT (Figure 4F,G,I). Movable finger with a series of three teeth— MM and MP of similar size and the MSM smaller than the others—, slightly retrolateral longitudinal carina composed by a row of barely visible granules, shallowed dorso-distal concavity and hook-shaped terminal teeth.

Chelicera-setose areas and stridulatory plate. Manus with abundant forked bristles on its dorsal and retrolateral areas (*rlm*) and hair of different size on its ventral surface (Figure 4G). The retrolateral fixed finger surface covered by fine tip setae of variable size (*rlf*); retrolateral movable cluster (*rlpc*) setae restricted to the proximal area of the finger and composed of few fine tip rigid setae with one flexible plumose seta (Figure 4G,H). The prolateral surface of the manus covered with several groups of setae, prodorsal proximal (*pdp*), promedial (*pm*) areas and promedial proximal cluster (*pmpc)* with tiny hair, proventral subdistal (*pvsd*) area with fine tip rigid setae arranged in sinuous row and other row of plumose setae on the proventral distal (*pvd*) side. Stridulatory plate subquadrangular located medially on the prolateral side and taking most of that side of the chelicera. Stridulatory apparatus with six well-developed ridges parallel to the dorsal and ventral surfaces (Figure 4J). Setose area of the movable finger ranging from the prolateral surface of the proximal area to the midpoint between MM and MSM, including prodorsal plumose setae (*mpd*), promedial flat-shape tip setae (*mpm*) and proventral flat-shape tip setae (*mpv*). Fondal area of movable finger with plumose setae (*mff*).

Chelicera-flagellum. Membranous, shaped as a leaf or spoon and divided into a distal and proximal lobe separated by a notch on its ventral edge (Figure 4K), and inserted in the chelicera through a movable cylindrical protrusion with an elliptical attachment structure placed at the level of the PFSM tooth. The insertion area is dorsally elevated and wider than the proximal lobe, but both approximately equal in length. The edge of the proximal lobe is serrated with short teeth.

Pedipalp. All segments covered by abundant short setae, with some medium-size setae in the proventral and retroventral surfaces, especially in the basitarsus and telotarsus. Femur divided in one short proximal and another long distal parts separated by a shallow and thin groove, the longer piece with two prodorsal distal elongated sensilla and two medial macrosetae on the proventral surface; tibia with three dorsal elongated sensilla, a middle one and two subdistal ones, and two long and thin setae; basitarsus and telotarsus with two dorsal elongated sensilla. 

Legs. Leg I coated with short and medium-sized setae like in the pedipalp. Femur with two ventral elongated sensilla; tibia with two dorsal—one proximal and one distal—, one prolateral and one retrolateral distal elongated sensilla; basitarsus with three distal elongated sensilla, one dorsal, one prolateral and one retrolateral; telotarsus without any claws, but with a multitude of specialized setae, some of which, with a bulb-shaped apex (Figure 4L). Walking legs with short and medium-sized setae like Leg I and pedipalp. Leg II, tibia with one dorsal trichobothrium-like sensillum on its distal area and two rows of thick distal spines on the ventral surface; basitarsus with three gross prolateral spines and six thick retrolateral spines in an irregular row; telotarsus segmented in two parts, the proximal with two thick prolateral spines and one retrolateral, and the distal with two prolateral and two retrolateral thick spines. Leg III, femur with one prolateral trichobothrium-like sensillum, tibia with one medial dorsal trichobothrium-like sensillum and three thick ventral spines in its distal portion; three prolateral, three dorsal and three retrolateral thick spines in basitarsus; telotarsus bi-segmented, two prolateral and one retrolateral thick spines in the proximal segment and two prolateral and two retrolateral spines in the distal one. Leg IV, femur with one proximal trichobothrium-like sensillum on its prolateral area; tibia with two dorsal and one prolateral elongated sensilla, one proventral row of six long size spines and two distal spines on the ventral area; basitarsus with one medial dorsal elongated sensilla, one row of four proventral spines and one row of two retroventral spines; telotarsus with five prolateral spines and four retrolateral spines. Opisthosoma as other body segments. Ctenidia on the genital and first post-genital sternite (Figure 4M) with the same appearance of forked setae, but somewhat larger and denser.

### 3.5. Description (Female)

Measurements in Table 2.

Color as male. Prosoma. Propeltidium subhexagonal, wider than longer (Table 2). Forked setae and hair varying in size across the entire surface (Figure 5A). Ocular characters as in the holotype (Table 2). Anterolateral propeltidal lobes separated by an incomplete deep groove. Mesopeltidium and metapeltidium rectangular with forked setae (Figure 5B). Coxae also covered by abundant forked setae. Chelicerae-dentition, processes, setose areas and stridulatory plate as described for the male. Pedipalps and legs as holotype. Ctenidia on the first post-genital sternite poorly visible or absent (Figure 5C).

### 3.6. Statistical Analyses

Multiple factor analysis showed that for both males and females, the two species were clearly separated in two distinct clusters with non-overlapping 95% confidence ellipses (Figure 6). Additionally, the MANOVAs were highly significant for both sexes (females: F_2,11_ = 61.4, *p* < 0.0001; males: F_2,13_ = 67.4, *p* < 0.0001). For females, the MFA explained 39.4% of the variance in its first dimension and 27.1% in the second. For males, the estimates were 49.2% and 17.8%, respectively, for each dimension. Concerning the contribution of each variable group, in females, the leg variables followed by the qualitative variable (color) and the propeltidium group contributed most to both dimensions (Table 3). The same was true for males, although in addition to color, the qualitative group of variables also included the presence of a skirt-like row of dense hair. However, the flagellum group contributed the most (17.1%) in the first (and by far the most important) dimension for males, while the propeltidium contributed more relative to the other structures than it did in females, with a maximum of 32% in the second dimension (Table 3).

### 3.7. Molecular Analyses

Molecular distances calculated by the Kimura two-parameter method [52] are given in Figure 7. This shows values of less than 0.005 among individuals of the same locality, which indicates a strong correspondence to geographical affinity. Distances among the different localities excluding Almería ranged from 0.0315 to 0.0573, whereas distance between individuals from Almería and those from the other localities were substantially higher, i.e., 0.1084–0.1207. The *G. dorsalis* sequence downloaded from GenBank showed a distance mean of 0.1202 for individuals from Almería, while this value was 0.0439 for individuals from the other localities. The distance values between the genus *Gluviopsis* (*G. nigrocinctus*) and the genus *Gluvia* (*G. dorsalis* and *G. brunnea* sp. nov.) were always greater than 0.2. Moreover, the distance values from other Solifugae families (*E. scopulatellus*-Eremobatidae family) were higher. The neighbor-joining (NJ) tree constructed with K2P distances of the different Solifugae specimens shows groups clearly separated by geographical localities and also shows the individuals from Almería in a distinct cluster, clearly separated from the other groups (Figure 8). The longer distance between individuals from Almería (*G. brunnea* sp. nov.) and the other specimens in the genus *Gluvia* is also conspicuous in the tree.

## 4. Discussion

### 4.1. Intraspecific Variation in Morphology and Recent History of the Genus

Overall, our morphological diagnoses and descriptive comparisons, along with the statistical and molecular DNA analyses, support the hypothesis that *G. brunnea* sp. nov. is a distinct, separate species from *G. dorsalis*. The morphologies of mature females of the two species are very similar, which, beyond the quantitative multidimensional statistical analysis (MFA, see below), can be differentiated by the characteristics outlined in the diagnosis section, i.e., eye disposition, the presence of hypertrophied macrosetae in coxa I and the color. On the contrary, the morphology of mature males shows several distinctive characteristics. The intraspecific variation observed in both species includes the number of fondal teeth and the number of ridges of the stridulatory plate. Females of *G. dorsalis* have seven or eight ridges and three or four profondal and retrofondal teeth (the specimen from Colmenar Viejo, Madrid, presents two profondal and two retrofondal teeth), and males bear between three and four profondal and retrofondal teeth and five to seven ridges. Meanwhile, the females of *G. brunnea* sp. nov. have seven to nine ridges, three or four profondal teeth and four or five retrofondal teeth, and males have two RFSP, one PFSM and seven or eight ridges in the stridulatory plate.

The genus *Gluvia* has been characterized by previous authors by the absence of ctenidia [20]; however, it is possible to see in both species a group of thicker setae than the setae that usually cover the second ventral sternites of the opisthosoma (Figure 9). These setae are more conspicuous on mature males than in females of *G. brunnea* sp. nov. and than in both sexes of *G. dorsalis*, in which the number of these structures is considerably lower (they are sometimes very difficult to see or absent in some specimens). In addition, most *G. dorsalis* males have a skirt-like row of dense hairs on the fourth post-genital sternite (Figure 9), which was not observed in any of the *G. brunnea* sp. nov. individuals. These structures have not been documented before. Apparently, in the latest reviews of the genus *Gluvia* [19,20], the authors did not give enough weight to these characteristics and the species *G. brunnea* sp. nov. went unnoticed, probably also as a result of not giving enough thought to the dental formulae and the flagellum morphology. Indeed, in their Figure 7, González-Moliné et al. (2008) clearly show the chelicerae of the *G. brunnea* sp. nov. mature male, to illustrate what they described as fixed finger abortive teeth in *G. dorsalis*. However, this figure shows several diagnostic characteristics used to distinguish between the two species, as discussed in the diagnostic section of the present work. It is highly likely that the photographed specimen was a sample from the localities of Almería (given that the authors included specimens from the province). Furthermore, the descriptions of other species from the Iberian Peninsula—[16], *G. minima* [14], *G. striolata* [55]— either fit with the latest description of *Gluvia dorsalis* (see González-Moliné et al. 2008 [20]) or the characteristics presented are inconclusive and erroneous—*G. dorsalis* var. *conquensis* [56]. When considering continuous intraspecific variation in the flagellum, all of the above taxa and some others (e.g., *G. chapmani* [16]) were designated as synonyms of *G. dorsalis* by Rambla and Barrientos in 1983 [19]. After a careful check of the above literature, we are confident that *G. brunnea* sp. nov. does not match any of the synonymies of [19], and thus it is a valid new species.

### 4.2. Specimens of Solifugae Captured in Mesovoid Shallow Substratum Stations

Mesovoid Shallow Substratum stations are a type of trap buried as deep as 0.9 m in the ground, and are ideal to sample fauna active in deep soil. We found two subadults of the new species in traps set in Sierra de María (northern Almería). To our knowledge, this is the first time that specimens of Solifugae have been found in this type of trap.

### 4.3. Multiple Factor Analysis Followed by MANOVA, a Successful Procedure to Separate New Species

We found that MFA successfully separated the two species when using both male and female data. Additionally, these analyses showed which group of characteristics contributed the most, with qualitative data (color and presence/absence of structures) and the propeltidium and leg biometrics contributing the most to MFA axis variance in females. In males, the flagellum group had also the strongest weight in the first dimension. We propose that the analytical framework that we present here may be superior to other approaches used to separate species. First, unlike principal component analysis (PCA) or discriminant analysis (DA), MFA allows for accommodating continuous, count and qualitative characteristics. Also, unlike DA, MFA can provide reliable results even when the number of cases is smaller than the number of variables. Additionally, in DA, the species are established as groups a priori and one tests which characteristics best discriminate between the species, making the implicit assumption that these species are valid entities. The approach of MFA is to include all traits and ordinate them in multidimensional space, and then map the corresponding centroids and ellipsoids of the hypothesized species (a posteriori). Finally, as we show here, extracting the dimension scores can be used to run a MANOVA to test whether there is significant separation of the hypothesized species across multidimensional space. At least in our case, the scores met all MANOVA assumptions, suggesting that the output of MFA is generally amenable to MANOVA. Alternatively, when assumptions are not met, a permutation test such as that in redundancy analysis [57] could be applied.

### 4.4. Molecular DNA Evidence

In general, in studies of arachnids, intraspecific COI distances range between 0 and less than 5% [30,58], while interspecific distances may range between 6.23 and 13% [28,58]. The genetic distances between *G. brunnea* sp. nov. and *G. dorsalis* (range 10.8–12.1%) confirm there are substantial genetic differences between the two taxa and allow for the recognition of the former taxon as a valid species, supporting the documented morphological differences above. Moreover, the distance values among individuals of the same locality (<0.5%) and among individuals of the different localities (3.1% to 5.7%) suggest substantial genetic divergence among populations but not enough to designate them as different species. Furthermore, in addition to the morphological characteristics, the distances between all individuals coming from localities other than Almería and of *G. dorsalis* (sequence from GenBank (MD038)) confirm that these individuals belong to *G. dorsalis* (distance 0.25 to 6.17%) and that all presumed *Gluvia* specimens included in the molecular analysis likely belong to this genus (distances < 12.1%). In summary, the distances obtained by the K2P method establish the intraspecific genetic distance as between 0 and 5.7% and the interspecific distance as from 10.8% to 12.07%. The distances between the genus *Gluviopsis* (*G. nigrocinctus*) and the genus *Gluvia* (*G. dorsalis* and *G. brunnea* sp. nov.) are over 20%, supporting the generic placement of these two taxa within the family. All these facts are well-established in the NJ tree (Figure 7).

## 5. Conclusions

We have used an integrative approach, including morphological descriptions, multivariate statistical analyses and molecular DNA analyses, to describe the second irrefutable species of Solifugae in the Iberian Peninsula, following the description of *G. dorsalis* by Latreille in 1817. We hope that approaches such as the one presented here encourage researchers to seek more species in this and other territories.

## Figures and Tables

**Figure 1 insects-15-00284-f001:**
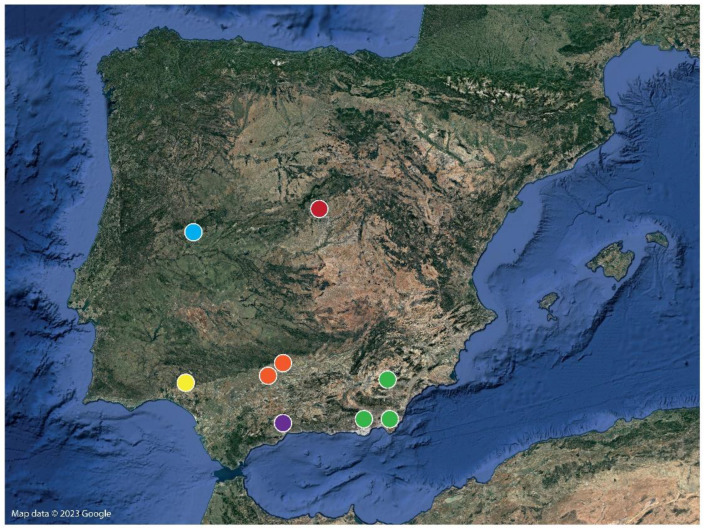
Areas where samples were collected. Blue, orange, yellow, red and purple dots (Cáceres, Córdoba, Huelva, Madrid and Málaga, respectively) indicate the collecting localities for *G. dorsalis.* Green dots (Almería) indicate the area where individuals of the new species *Gluvia brunnea* sp. nov. were collected.

**Figure 2 insects-15-00284-f002:**
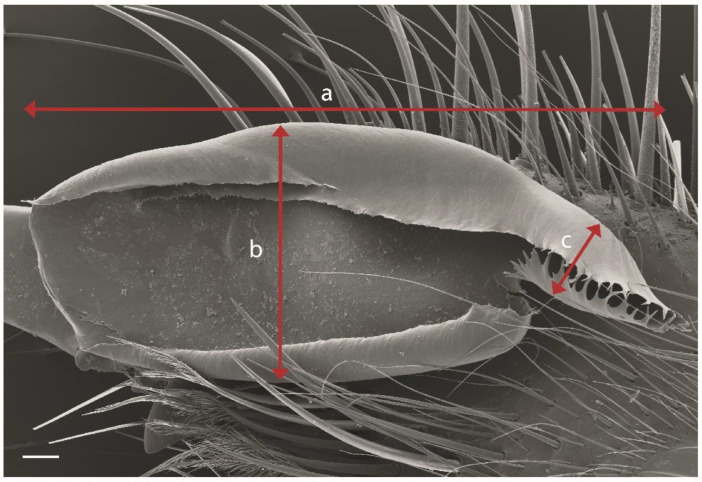
Image of flagellum of *G. dorsalis* adult male. Length (a), width of distal lobe (b) and width of proximal lobe (c). Scale bar: 0.1 mm.

**Figure 3 insects-15-00284-f003:**
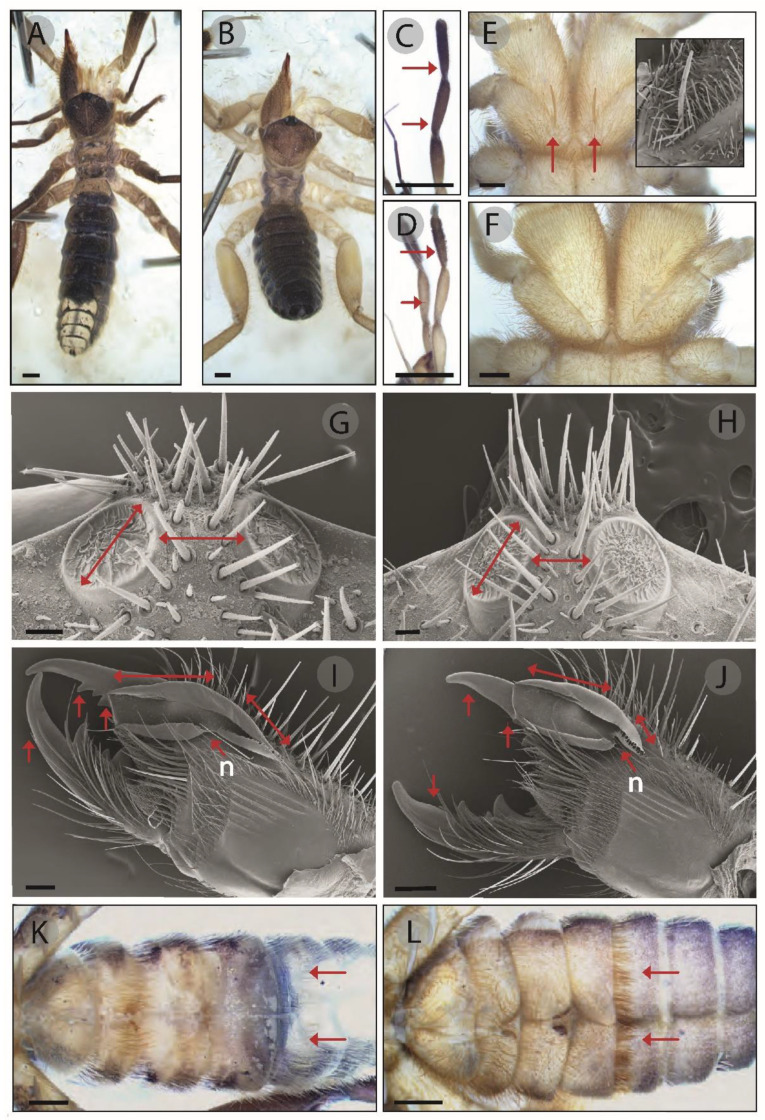
Mature males. *G. brunnea* sp. nov (**A**,**C**,**E**,**G**,**I**,**K**) and *G. dorsalis* (**B**,**D**,**F**,**H**,**J**,**L**). (**A**,**B**) Body color. (**C**,**D**) Palp coloration. (**E**,**F**) Coxae I: arrows indicate the macrosetae present in *G. brunnea* sp. nov. (**G**,**H**) Ocular lobe: arrows indicate the different proportions in the lengths of the eyes and the interocular distances. (**I**,**J**) Chelicerae: single-tip arrows indicate the shapes of movable finger mucrons, the positions of distal teeth and the notch “n”; double-tip arrows indicate the proportional differences in the flagellum. (**K**,**L**) Ventral view of the opisthosoma: the arrows show the presence of the skirt-like row of hair in *G. dorsalis*, which is absent from *G. brunnea* sp. nov. Scale bars: 1 mm (**A**,**B**,**K**,**L**), 0.5 mm (**C**,**D**,**E**,**F**,**J**), 0.2 mm (**I**), 0.1 mm (**G**,**H**).

**Figure 4 insects-15-00284-f004:**
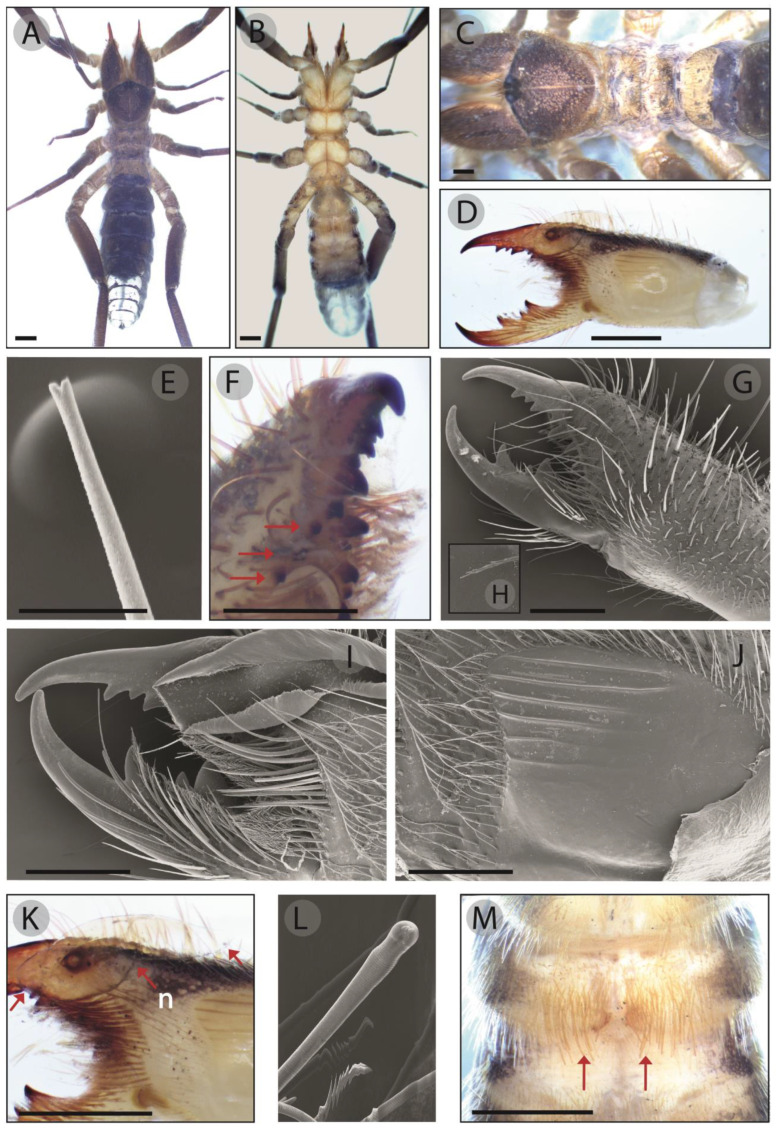
Adult male of *G. brunnea* sp. nov. Holotype (**A**–**D**,**F**,**K**,**M**). (**A**) Habitus, dorsal view. (**B**) Habitus, ventral view. (**C**) Propeltidium, mesopeltidium and metapeltidium. (**D**) Prolateral side of chelicerae. (**E**) Forked tip of a seta. (**F**) Ventral view of fixed finger, where arrows show the retrolateral fondal teeth. (**G**) Retrolateral view of chelicera. (**H**) Tip of retroventral plumose seta. (**I**) Distal area of chelicera. (**J**) Stridulatory plate of an adult male. (**K**) Flagellum of holotype, where arrows indicate the distal and proximal lobes and the notch (n). (**L**). Specialized seta of leg I with bulb-shaped tip. (**M**). Holotype ctenidia. Scale bars: 1 mm (**A**–**C**,**G**,**K**,**L**), 0.5 mm (**F**,**I**,**J**), 0.05 mm (**F**).

**Figure 5 insects-15-00284-f005:**
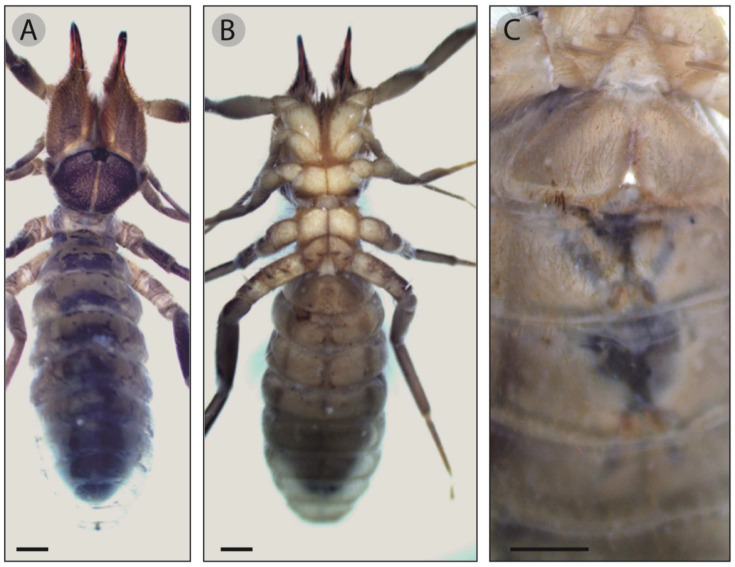
Adult female. (**A**) Dorsal view. (**B**) Ventral view. (**C**) Genital and post-genital sternites. Scale bars: 1 mm.

**Figure 6 insects-15-00284-f006:**
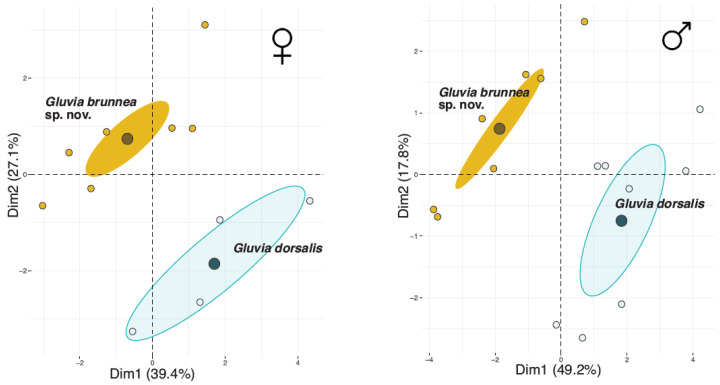
Results of multiple factor analyses (MFAs) for each sex. (**Left panel**) Females. (**Right panel**) Males. Brown dots and 95% confidence ellipses, *G. brunnea* sp. nov.; blue dots and 95% confidence ellipses, *G. dorsalis*. Larger circles correspond to the centroids of each species. The species identity was included as a supplementary categorical variable for analysis, and thus it had no effect on the ordination.

**Figure 7 insects-15-00284-f007:**
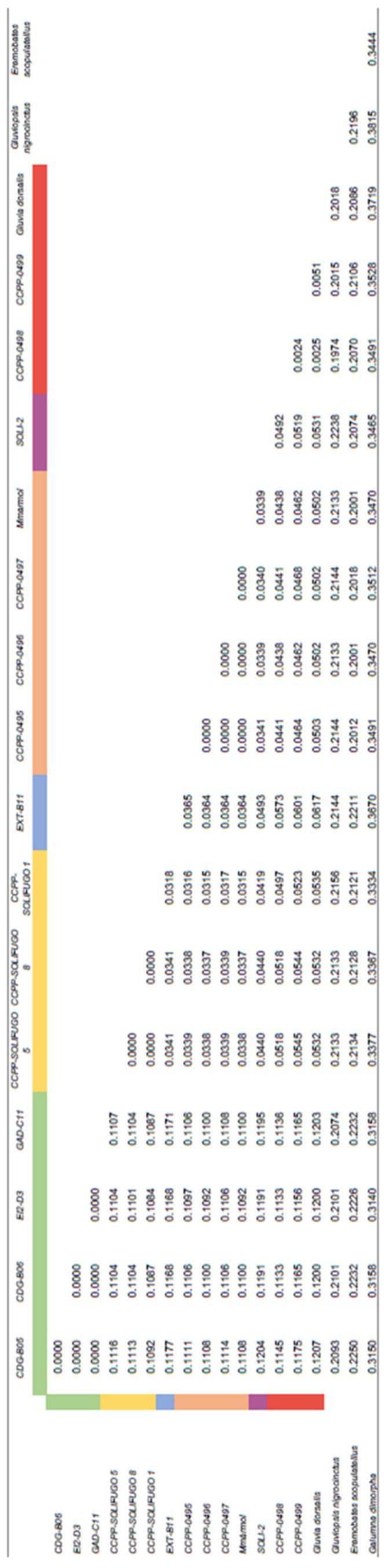
Distance matrix calculated by the Kimura 2-parameter method. The colors show groups of individuals with a distance < 0.01, corresponding to the same collection localities.

**Figure 8 insects-15-00284-f008:**
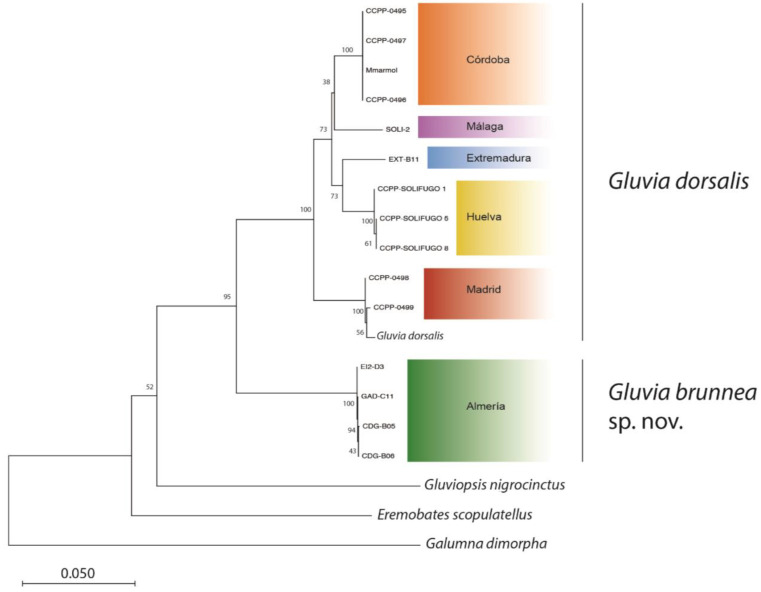
NJ tree. Bootstrap support values are shown above the branches.

**Figure 9 insects-15-00284-f009:**
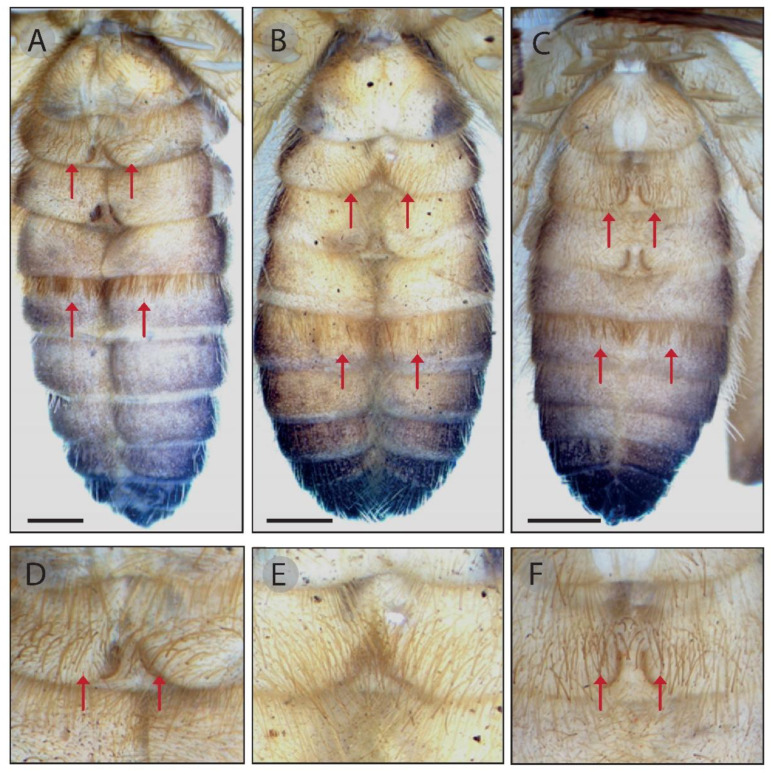
Ventral sternites of three adult males of *Gluvia dorsalis* from different localities, where arrows indicate the hairy skirt and the zone where the ctenidia should be located. (**A**,**D**) Córdoba. (**B**,**E**) Huelva. (**C**,**F**) Mature Madrid. ((**D**–**F**) Second ventral sternite details.)

**Table 1 insects-15-00284-t001:** Specimens used for molecular analysis in the present study.

Taxon Name	Locality	Data	Code	GenBank Access Number
*Gluvia brunnea* sp.nov.	Cabo de Gata, Almería	19 May 2022	CDG-B05	PP229911
*Gluvia brunnea* sp.nov.	Cabo de Gata, Almería	11 July 2022	CDG-B06	PP229912
*Gluvia brunnea* sp.nov.	Sierra de Gádor, Almería	10 July 2021	EI2-D3	PP229913
*Gluvia brunnea* sp.nov.	Sierra de Gádor, Almería	14 August 2023	GAD-C11	PP229914
*Gluvia dorsalis*	Huelva	01 June 2018	CCPP-SOLIFUGO 5	PP229915
*Gluvia dorsalis*	Huelva	18 June	CCPP-SOLIFUGO 8	PP229916
*Gluvia dorsalis*	Huelva	18 June	CCPP-SOLIFUGO 1	PP229917
*Gluvia dorsalis*	Sierra de Gata, Cáceres	30 June 2008	EXT-B11	PP229918
*Gluvia dorsalis*	Alcolea, Córdoba	28 July 2023	CCPP-0495	PP229919
*Gluvia dorsalis*	Alcolea, Córdoba	28 July 2023	CCPP-0496	PP229920
*Gluvia dorsalis*	Alcolea, Córdoba	28 July 2023	CCPP-0497	PP229921
*Gluvia dorsalis*	El Patriarca, Córdoba	29 April 2023	Mmármol	PP229922
*Gluvia dorsalis*	Málaga ciudad	07 August 2001	SOLI-2	PP229923
*Gluvia dorsalis*	Colmenar Viejo, Madrid	22 July 2023	CCPP-0498	PP229924
*Gluvia dorsalis*	Colmenar Viejo, Madrid	22 July 2023	CCPP-0499	PP229925
*Gluvia dorsalis*	-	-	-	MD038
*Gluviopsis nigrocinctus*	-	-	-	SOL162
*Eremobates scopulatellus*	-	-	-	DMNS:ZA19206
*Galumna dimorpha*	Cabo de Gata, Almería	05 February 2021	-	PP228019

**Table 2 insects-15-00284-t002:** Measurements.

		Holotype	Males	Paratype Female	Females
	Body length	12.95	8.31–15.18	20.65	11.12–20.65
Chelicerae	Dorsal width	1.24	0.89–1.32	2.55	1.36–2.55
	Lateral width	1.26	0.96–1.48	2.83	1.47–2.83
	Lateral length	3.5	3.08–4.11	6.72	4.67–6.72
	Flagellum length	1.57	1.17–1.71	-	-
	Flagellum distal lobe width	0.54	0.42–0.65	-	-
	Flagellum proximal lobe width	0.33	0.29–0.37	-	-
Propeltidium	Area	4.27	2.67–5.67	15.97	5.44–15.97
	Flatness ratio	0.84	0.8–0.87	0.66	0.62–0.71
	Width	2.76	2.16–3.17	5.77	3.41–5.77
	Length	2.32	1.79–2.64	3.91	2.37–3.91
	Ocular tubercle (area)	0.31	0.19–0.42	0.51	0.27–0.51
	Interocular	0.24	0.15–0.29	0.38	0.17–0.38
	Eye length	0.3	0.21–0.33	0.31	0.24–0.36
Pedipalp	Telotarsus	0.96	0.72–0.99	0.82	0.68–1.1
	Basitarsus	2.79	2.29–3.28	3.54	2.63–3.54
	Tibia	3.62	2.88–4.29	4.35	2.86–4.35
	Femur	4.12	3.04–4.68	4.53	2.82–4.53
	TOTAL	11.49	8.95–13.23	13.24	9.01–13.24
Leg I	Telotarsus	1.45	0.84–1.45	1.46	1.02–1.87
	Basitarsus	1.52	1.13–2.18	2.38	1.16–2.38
	Tibia	2.81	2.36–3.67	3.91	2.17–3.91
	Femur	2.73	1.61–2.78	2.74	1.5–2.86
	TOTAL	8.51	6.45–10.05	10.49	6.33–10.49
Leg II	Claw	0.69	0.45–0.81	0.75	0.57–0.8
	Telotarsus	0.71	0.57–0.9	1.16	0.6–1.16
	Basitarsus	1.15	1.02–1.57	1.84	1.13–1.84
	Tibia	1.55	1.43–1.96	2.4	1.53–2.4
	Femur	1.86	1.21–2.26	1.89	1.27–1.91
	TOTAL	5.96	4.68–7.5	8.04	5.14–8.04
Leg III	Claw	0.86	0.55–0.93	0.72	0.69–0.96
	Telotarsus	0.9	0.65–0.99	1.33	0.71–1.33
	Basitarsus	1.69	1.26–1.96	2.48	1.38–2.48
	Tibia	2.19	1.68–3.07	3.18	1.78–3.18
	Femur	2.78	1.82–3.18	3.02	1.93–3.02
	TOTAL	8.42	6.6–10.03	10.73	6.75–10.73
Leg IV	Claw	1.42	0.99–1.38	1.26	0.91–1.31
	Telotarsus	1.74	1.47–2.09	2.23	1.17–2.23
	Basitarsus	3.71	2.56–4.24	4.02	2.54–4.02
	Tibia	5.02	3.46–5.71	5.68	3.31–5.68
	Femur	4.63	3.34–5.5	5.51	2.77–5.51
	TOTAL	16.2	10.94–18.92	18.7	10.8–18.7

**Table 3 insects-15-00284-t003:** Contribution (%) of each character group to the first 2 dimensions of the MFA.

**Females**		
*Group*	*Dim.1*	*Dim.2*
qualitative	10.00	23.94
propeltidium	16.59	28.53
eyes	22.09	0.80
palps	21.63	6.83
legs	27.15	4.31
chelicerae	1.31	19.61
stridulatory organ	1.24	15.97
**Males**		
*Group*	*Dim.1*	*Dim.2*
qualitative	11.48	15.36
propeltidium	11.72	32.44
eyes	11.40	10.03
palps	12.86	8.67
legs	14.59	9.09
chelicerae	11.37	12.39
stridulatory organ	9.51	3.09
flagellum	17.07	8.93

## Data Availability

https://doi.org/10.20350/digitalCSIC/16184.

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
