# Peer review of "More Than 200 Years Later: Gluvia brunnea sp. nov. (Solifugae, Daesiidae), a Second Species of Camel Spider from the Iberian Peninsula"

_insects, 2024, doi:10.3390/insects15040284_

Round 1
Reviewer 1 Report
Comments and Suggestions for Authors
Review of insects-2900489
Title: More than 200 years later: Gluvia brunnea sp. nov. (Solifugae, 2 Daesiidae), a second species of camel spider from the Iberian 3 Peninsula
General Comments:
This is a very nice paper in which authors describe a new species of solifuge in the genus Gluvia and use multiple lines of evidence including creative statistical analyses and descriptions and identification of more traditional morphological diagnostic characters. The changes highlighted in the accompanying review pdf as well as detailed below are largely minor corrections and suggested changes. I would, however, suggest moving the taxonomic description after the presentation of the statistical results.
Minor corrections (mostly typos): These are highlighted in the review copy of the pdf in yellow
Abstract, line 17, change “de” to “a”
Lntroduction, line 33: change “systematic” to “systematics”
Line 36: correct spelling of cheliceral
p. 2, line 49: In what way is the COI sequence useful in supporting differences in morphological characters? This does not make sense. It may be useful in supporting the separation of two populations of solifuges into two separate species BASED upon morphologically distinct differences. But it doesn’t make sense to argue the other way around.
Line 50: do not capitalize “arachnid”
Line 51: the order name is Opiliones not Oplionida
Line 66: change to “kind of study” and “type of station”
Line 69: delete the leading “-“ in front of 0.9 m
Line 70: change wording to: “bait is placed at the bottom inside the cylinder”
Line 72 (and elsewhere – search for this): alcohol concentration should be followed by % not by ° [96% alcohol not 96° alcohol; 70% alcohol, not 70°]
Throughout the methods, use past tense not present tense as indicated below:
Line 73 change to “Almería, were collected”
Line 75: change to “entomofauna was affected in a burned [not burnt] area
Line 76: change to “Each trap consisted of a 7.5 cm… and a 200 ml plastic container [not glass]”
Line 77: change “glasses” to “containers”
Line 88: change to “set of specimens came from”
Line 91: change to “traps were set”
Line 102: change the wording under Scanning Electron Microscope section, 1st sentence, to “SEM (Scanning Electron Microscope) was used to visualize six specimens; two pairs…..of the new species.”
Line 111: correct spelling of “flatness”
Line 112: I have no idea, even after looking at the Figure, what is meant by “flag flagellum width” and if you mean the width of the flagellum, shouldn’t you have measured this PRIOR to visualizing with SEM since drying the chelicera will, inevitably, reduce the size and cause additional curling of the delicate flagellum. You at least need to better explain what “flag flagellum width” means.
Lines 133-135: The highlighted sentence is a run-on sentence and should be re-worded.
Line 140: correct the spelling of “analysis”
Line 141: correct the spelling of “flattening”
Line 170: 430 pb should be 430 bp
Line 171: Capitalize the name of the order Solifugae
Lines 172-173: change to “The COI fragment was amplified by PCR”
Table 1 – the Eremobates scopulatellus specimen # was DMNS ZA.19206 NOT DMS – please correct in table and in text.
Line 183: Correct the spelling of the family Daesiidae
Line 186: change to “was chosen”
Line 189: correct spelling of GENIOUS
Line 223: change “alive” to “living”
It would be preferable to move the results of the statistical analysis before the Taxonomy section.
In the Taxonomy section (or before in Methods) please define the abbreviations of the various collections (e.g., CECOUAL, MNCN, etc).
Figures 3A and B (the habitus photos) are not very sharp or good quality – is it possible to re-take these images?
In the Figure legend of Figure 3, don’t you mean that Figure 3E shows the “macrosetae present in G. brunnea sp. Nov.”? You said G. dorsalis sp. nov.
Lines 236 and throughout this section, please reference the exact figure in Fig 3 – i.e. “Fig. 3A” or “Fig 3B” – don’t just reference Fig. 3.
Lines 240-241: Change the wording in the sentence beginning “Contrary than…” Maybe instead say that “The interocular distance in [whichever] species is [then provide the comparative value – which I don’t understand the way you have worded it.]”
In Table 2, “flagelo” should be spelled “flagellum” and by “Flattenes” I think you mean “Flatness”
In Fig 4K, would it be possible to re-take this image against a darker background so it is easier to visualize the shape of the flagellum? That would be helpful.
Line 297: “composed by few” should be changed to “composed of few”
Line 302: Change “disposed in the middle of prolateral side and taking the most of the piece” to “located medially on the prolateral side and taking up most of that side of the chelicera.”
Line 323 begins with a lonely comma that should be deleted.
Lines 321-342: You describe trichobothria on the legs. It is thought that Solifugae (the entire order) do not have trichobothria. Are you certain that what you describe as trichobothria truly are these specialized setae? If they do not sit in a perfectly circular depressed base, then they are likely just long setae and should not be referred to as trichobothria. If they ARE trichobothria, it would be useful for you to include a magnified image of the base of one.
If Fig 5C is supposed to show the genital operculum of the female, it should be re-taken. The habitus photos are also poor quality. In addition, if Fig 5C shows the genital operculum, it does not look like a sclerotized opercular plate to me.
Also, did you see appreciable differences between the opercula of the two species? If so, you should include this in the diagnosis for the new species.
In Table 3, there is no reason to abbreviate the traits (e.g. “chel,” “flage,” “qual” [not even sure what qual means]. Spell out the traits.
Line 382: delete “the” preceding GenBank
Line 386: “relatively” should be “relative” to
Lines 406-407: change “shows several diagnosis characters” to “shows several diagnostic characters”
Line 409: “has” should be “have”
Line 469: please change “confirm” to “support”
Please see accompanying highlighted pdf to see where changes should be made.

Minor corrections are necessary to English usage - these are detailed in my comments to the authors and highlighted in the pdf review version of the manuscript.
Author Response
Dear Reviewer,
Thank you for your helpful comments and edits, which have greatly improved the quality of the ms. We apologize for the many mistakes in English usage that went unnoticed. In this version, all authors have more carefully read the submitted version, and the sènior author has throughly edited the English. We hope that the ms is now ready for publication in Insects.
Best regards,
Cristian.
Reviewer 2 Report
Comments and Suggestions for Authors
Please see the attachment.

I suggest a moderate editing of english.
Author Response
Dear Reviewer,
We are grateful with your work and your commets and suggestions have improved the manuscript. We apologize for the many mistakes in English usage that went unnoticed. In this version, all authors have more carefully read the submitted version, and the senior author has throughly edited the English. We hope that the ms is now ready for publication in Insects.
There are a small problem, we don´t know which reviewer did each comment in the manuscript.
Below we clarify the comments which need some explanation:
1. Fondal is the name used for basal teeth in the literature as Bird et al 2015.
2. Discussion section. We feel that the correct place to comment on the morphological variations of the two species is in this section, as we compare how each species varies.
3. Discussion section. In the last revision of Gluvia (Gonzalez-Moliné et al 2008), the authors studied and photograpied specimens of G. brunnea sp. nov. but they didn´t take into account the differences in the chelicera and flagellum morphology between two species. According to them, these differences are due to anomalies. They overlooked the new species.
3. Discussion section. We add a new small paragraph about the samples collected with Undergroudn Sample Stations.
4. Discussion section. The manuscript is meaningless without statistical and molecular analyses, which support and complement the morphological study.
Thank you again for your work.
Best regards,
Cristian.